# Novel bioinformatic methods and machine learning approaches reveal candidate biomarkers of the intensity and timing of past exposure to *Plasmodium falciparum*

Sophie Bérubé[1]*, Tamaki Kobayashi[2], Douglas E. Norris[3], Ingo Ruczinski[1], William J. Moss[2,3], Amy Wesolowski[2], Thomas A. Louis[1]

1 Department of Biostatistics, Johns Hopkins Bloomberg School of Public Health, Baltimore, MD, United States of America, 2 Department of Epidemiology, Johns Hopkins Bloomberg School of Public Health, Baltimore, MD, United States of America, 3 Department of Molecular Microbiology and Immunology, Johns Hopkins Bloomberg School of Public Health, Baltimore, MD, United States of America

* sberube3@jhmi.edu

**Data Availability Statement:** Relevant data and code are available on a public repository,

## Abstract

Accurately quantifying the burden of malaria over time is an important goal of malaria surveillance efforts and can enable effective targeting and evaluation of interventions. Malaria surveillance methods capture active or recent infections which poses several challenges to achieving malaria surveillance goals. In high transmission settings, asymptomatic infections are common and therefore accurate measurement of malaria burden demands active surveillance; in low transmission regions where infections are rare accurate surveillance requires sampling large subsets of the population; and in any context monitoring malaria burden over time necessitates serial sampling. Antibody responses to *Plasmodium falciparum* parasites persist after infection and therefore measuring antibodies has the potential to overcome several of the current obstacles to accurate malaria surveillance. Identifying which antibody responses are markers of the timing and intensity of past exposure to *P. falciparum* remains challenging, particularly among adults who tend to be re-exposed multiple times over the course of their lifetime and therefore have similarly high antibody responses to many *Plasmodium* antigens. A previous analysis of 479 serum samples from individuals in three regions in southern Africa with different historical levels of *P. falciparum* malaria transmission (high, intermediate, and low) revealed regional differences in antibody responses to *P. falciparum* antigens among children under 5 years of age. Using a novel bioinformatic pipeline optimized for protein microarrays that minimizes between-sample technical variation, we used antibody responses to *Plasmodium* antigens as predictors in random forest models to classify samples from adults into these three regions of differing historical malaria transmission with high accuracy (AUC = 0.99). Many of the most important antigens for classification in these models do not overlap with previously published results and are therefore novel candidate markers for the timing and intensity of past exposure to *P. falciparum*. Measuring antibody responses to these antigens could lead to improved malaria surveillance.

information and links are in the the Methods section of the text. Repository link is: https://github.com/sberube3/Antigen_classification_code.

**Funding:** NIH-NIAID, U19-AI089680 Career Award at the Scientific Interface from the Burroughs Wellcome Fund awarded to Amy Wesolowski.

**Competing interests:** The authors have declared that no competing interests exist.

## 1 Introduction

Accurately quantifying the burden of malaria is an important component of malaria control and elimination efforts [1] allowing more targeted interventions to be deployed and evaluated [2]. Currently, common malaria surveillance methods include measuring incidence or prevalence of malaria infections based on active or recent infections using diagnostic tests (e.g., microscopy, rapid diagnostic tests, or polymerase chain reaction). However, there are challenges associated with using these methods. In low transmission settings, incident infections are rare, and in high transmission areas asymptomatic infections are common; both factors make detecting active or recent infections resource intensive [3–5]. Additionally, surveillance that detects active or recent infections provides estimates of malaria burden at single point in time. While this information is useful, collecting these data across multiple time points is essential to answer important scientific questions such as how malaria transmission changes over time, following interventions, and seasonally. With existing methods, surveillance over time requires sampling systems like household surveys or reactive case detection to be sustained for long periods [6, 7]. Antibody responses to *Plasmodium falciparum* infection have the potential to overcome several of these challenges. Antibodies persist for extended periods after infection and can provide insight into an individual's or a population's infection history without needing to capture active infections. Serological data can also provide longitudinal information about malaria burden with a cross-sectional sample, which requires substantially fewer resources to obtain [8, 9].

The ability to use antibody responses to *P. falciparum* infections for surveillance purposes is complicated by several factors including: multiple developmental phases of the parasite in humans, each with different expressed genes that produce a wide range of antigens; the presence of multiple, genetically distinct clones in a single infection, possibly eliciting different antibody responses; and reinfection events which are common over short time scales in high transmission settings [10–12]. Despite these challenges, studies have illuminated many aspects of the human antibody response to *P. falciparum* infections, including correlates of protective immunity, possible vaccine targets [13, 14], and markers of recent infection [15, 16]. However, such studies have largely been successful in children under the age of 5 years, and expanding these studies to broader age groups is complicated by the fact that older individuals in malaria endemic settings tend to be re-exposed and re-infected multiple times throughout their life. These infection events give adults across regions of differing transmission similarly high reactivity to *P. falciparum* antigens [15, 17, 18], and this makes distinguishing antibody responses based on differences in past exposure difficult. To make inferences about past malaria transmission levels over longer time frames, an important surveillance goal, antibody responses from adults who have a longer history of exposure are essential. Additionally, adults make up the majority of the asymptomatic reservoir which is thought to be disproportionately responsible for onward transmission [19, 20]. Therefore, serology based surveillance methods based on antibody responses from adults have the potential to better characterize the asymptomatic reservoir, and inform more effective interventions.

To expand the search for markers of the intensity and timing of past exposure to *P. falciparum* in adults, we used novel bioinformatic tools to analyze protein microarray data measuring antibody responses to 500 *P. falciparum* and 500 *P. vivax* antigens in 479 serum samples from individuals aged 0–86 years in three regions across Zambia and Zimbabwe [17]. These three regions span low, intermediate, and high historical malaria transmission levels [21, 22]. If antibody responses to certain *Plasmodium* antigens predict an individual's region of residence with high accuracy across multiple age groups, these could be the result of differences in the frequency of past exposure over decades. Frequency of exposure is, in turn, highly dependent

on the intensity of transmission in this timeframe. Additionally, the probability of recent exposure is directly correlated to the level of transmission and, therefore, antibody responses that are relatively constant across regions of differing transmission levels are unlikely to be markers of recent exposure. Consequently, antibody responses that differentiate individuals residing in regions with different historical malaria transmission could be long-lasting markers of the intensity and timing of past exposure to *P. falciparum*. Measuring these antibodies could overcome the current challenges associated with malaria surveillance.

A previous analysis of these data published by Kobayashi et al. [17] revealed a correlation between region of origin and antibody responses to a subset of 30 highly seroreactive *P. falciparum* antigens among children under 5 years of age, but little correlation was observed among older individuals. Here, we conduct an analysis of these data with a novel bioinformatic pipeline [23, 24] that ranks antibody responses to each antigen within each sample thus providing a measure of antibody levels to each antigen that is relative to that individual's responses to other *Plasmodium* antigens. This relative measure minimizes the effects of between-sample technical variation, which tends to be high in protein microarray data [23, 25]. Minimizing this variation makes differentiating adult antibody responses, which tend to be uniformly high across all transmission settings, more feasible. The within-sample antigen ranks are then used as predictors in a random forest model [26] that classifies samples into their region of origin. Using subsets of between 20 and 260 antigens, we classify all samples, including those from adults, into their region of origin with high accuracy. The random forest approach generally out-performs parametric multivariate modeling (e.g., logistic regression), particularly when many antigens were used to predict the location of origin of a sample. This superiority suggests that between-antigen interactions play an important role in using antibody responses to differentiate exposure to varying levels of malaria transmission. A closer analysis of the antigens most important for classification reveals some overlap with previously identified markers of recent exposure, however, several identified antigens appear to be novel markers of the intensity and timing of past exposure and so, potentially useful tools for malaria surveillance.

## 2 Materials and methods

### 2.1 Study populations and sites

A total of 479 samples from Choma District, Zambia, Nchelenge District, Zambia, and Mutasa District, Zimbabwe were probed with protein microarrays containing 500 *P. falciparum* and 500 *P. vivax* antigens. Of these, 414 samples were collected in serial, cross-sectional, community-based surveys in 2015 across all three sites, including at health centers and 65 samples from 13 individuals residing in Choma District, Zambia were collected from 2013–2015 in a longitudinal study. Across the three locations, the proportion of samples positive for *P. falciparum* by rapid diagnostic test (RDT) at the time of sampling, was between 3 and 35% and the proportion of individuals who were febrile (temperature $\geq$ 38°C) at the time of sampling ranged from 5 to 21%. Additionally, ages of individuals at the time of sampling were comparable across all three sites and spanned a wide range (0–86 years) (Table 1) [17]. These were nearly identical to the set of samples analyzed in Kobayashi et al. [17], with the exception of 34 samples from Mutasa district which we included in our analysis but were not included in Kobayashi et al. [17]'s analysis.

Although the characteristics of individuals across the three sites were similar, the levels of historical malaria transmission differed greatly. Table 1 shows the average weekly number of RDT positive cases at health centers in each of the three regions from 2012–2016. These numbers revealed a consistently low prevalence of malaria in Choma District, Zambia, a consistently high prevalence of malaria in Nchelenge District, Zambia and an intermediate level of

**Table 1. Study population and site characteristics.**

| Site | Choma District, Zambia | Nchelenge District, Zambia | Mutasa District, Zimbabwe |
|---|---|---|---|
| Number of Samples | N = 166 | N = 122 | N = 190 |
| Mean Age (Range) in years | 32.8 (1–83) | 23 (1–65) | 15 (0–86) |
| % Female | 43 | 61 | 56 |
| Average Weekly RDT Positive Cases at Health Centers | | | |
| 2012 | 1.0 | 100.0 | 24.4 |
| 2013 | 0.7 | 98.3 | 28.8 |
| 2014 | 1.3 | 146.0 | 24.2 |
| 2015 | 0.9 | 115.0 | 6.9 |
| 2016 | 3.1 | 74.0 | 8.4 |
| % RDT Positive Samples | 21 | 35 | 3 |
| % Febrile at time of Sampling | 21 | 13 | 5 |

transmission from 2012–2014 followed by a sharp decline in 2015 in Mutasa District, Zimbabwe, concordant with when the sampling for this study took place. These figures were compiled from data published in Moss et al. and Mharakurwa et al. [21, 22], and are representative of malaria transmission levels in these regions from 2012–2016. All cases of malaria in these regions were caused by *P. falciparum* and *P. vivax* is not known to have been present in any of these three areas from 2012–2016, or currently.

All human samples used in this analysis were collected with protocols approved by the Johns Hopkins Bloomberg School of Public Health Institutional Review Board—IRB 3467. Additionally, approvals from the Tropical Diseases Research Center Ethics Review Committee (TDRC/ERC/2010/14/11), the Biomedical Research and Training Institute Institutional Review Board (AP102/11), and the Medical Research Council of Zimbabwe (MRCZ/A/1625), were obtained. Written consent was obtained from all participants in this study.

## 2.2 Pre-processing measurements from protein microarrays

Output from the pre-processing pipeline described in Bérubé et al. [23] was fed into a Bayesian model that produces full posterior distributions of the true, underlying protein signal. Briefly, for pre-processing, the ratio of observed foreground signal to background signal at each probe or antigen ($Y'$) was log transformed. Then, using a linear model described by Sboner et al. [25], array and subarray effects were estimated and subtracted from $\log(Y')$ to produce $\tilde{Y}$. Finally, these values are standardized using array specific sample means and sample standard deviations of control probes to produce $Y$, the input for the Bayesian model.

We denote measurements from microarrays $Y_{i,p}$ with the following subscripts:

- array number $i \in \{1, \ldots, I\}$, since each array corresponds to a single sample, $I = 479$ in this study.

- Antigens $p \in \{1, \ldots, P\}$, in this study $P = 1038$, each $p$ is a *Plasmodium* antigen on the array.

As described in Bérubé et al. [24], the Bayesian model used the transformed data, $Y$, to produce the full posterior distribution of the true underlying signal, denoted by $S$. We assumed that true underlying fluorescence intensity resulting from specific protein binding between the proteins in the sample and the probes on the array, $S$, adds to a term $e$, which represents the technical variation or biological variation that is not the result of differences in exposure to *P. falciparum* remaining in the measurements after the pre-processing pipeline [23], to produce the normalized observation $Y$. We analyzed control probes on the array to estimate the error

distribution *e*, and ultimately produced an estimate of true underlying signal given observed and pre-processed signal, or *S*|*Y*, for each antigen on the array; this estimate is a full posterior distribution. The full hierarchical Bayesian model describing this relationship is described in Bérubé et al. [24] and a simplified version is presented in the S1 Text.

## 2.3 Ranking pre-processed measurements

We computed the ranks of the true *S* at each antigen *i* on the array via:

$$T_{i,p}(S_{i,p}) = \text{rank of } S_{i,p} \text{ in array } i := \sum_{k=1}^{P} I_{\{S_{i,p} \geq S_{i,k}\}} \tag{1}$$

so that the smallest *S* had rank 1 and the largest had rank *P*. The estimator that minimizes squared error loss is the posterior mean:

$$\bar{T}_{i,p}(Y) = E_{S|Y}[T_{i,p}] = \sum_{k=1}^{P} pr(S_{i,p} \geq S_{i,k}) \tag{2}$$

We use burned in Markov Chain Monte Carlo (MCMC) draws [27] to compute this quantity by computing the rank of each protein in each array for each MCMC draw, and then averaging those ranks across the MCMC draws, a complete ranking procedure is described in Bérubé et al. [24].

Ranks used to perform this analysis and relevant metadata can be found at https://github.com/sberube3/Antigen_classification_code.

## 2.4 Random forest models

**2.4.1 Antigen selection.** We first fit a random forest model using the R package `randomForest` [28] using as predictors the ranks $\bar{T}_{i,p}$ of all 1038 active probes (500 *P. falciparum* or *P. vivax* antigens expressed in *E. coli* and 38 purified *P. falciparum* antigens) on all 479 arrays spotted with samples from individuals residing in Zambia and Zimbabwe (described in Section 2.1). The random forest classified samples into their region of origin (Choma District, Zambia; Nchelenge District, Zambia; and Mutasa District, Zimbabwe). We used the mean decrease in Gini index to determining variable importance. The top 20, 30, 65, 130, and 260 antigens were therefore those with the greatest mean decrease in Gini index as computed from this original model. After considering models fit to all 479 samples with the top 20, 30, 65, 130, and 260 antigens, we also considered models fit to subsets of samples using the top 20, 30, 65, 130, and 260 antigens as predictors. These subsets included children and adults (using a cutoff age of 5 years, and a cutoff of 15 years), and samples from individuals who were negative by rapid diagnostic test (RDT) at the time of sampling. For all random forest models we used a prior distribution with equal weights for each of the three classes (regions), the number of candidate variables selected randomly at each split was taken to be $\sqrt{p}$, where *p* is the number of predictors in the model, and the number of trees to grow was set at 500.

**2.4.2 Assessing model performance.** Models fit with either subsets of antigens as predictors or subsets of samples are evaluated using both out of bag (OOB) error [26, 28], and multiclass AUC with corresponding 95% confidence intervals are computed using the R package multiRoc [29]. Specifically the global multiclass AUC is computed using the macro-average method of class-specific AUC as described by Hand and Till [30]. In addition to the OOB error and AUC computed by fitting random forest models to all 479 samples, we also evaluate the performance of models with a $\frac{2}{3} - \frac{1}{3}$ train-test split, and compute a test error, as well as an

AUC with the reserved $\frac{1}{3}$ test data. All relevant R code to execute model fitting and checks can be found at https://github.com/sberube3/Antigen_classification_code.

## 2.5 Polytomous logistic regression models

We assessed the performance of following polytomous logistic regression models in classifying samples into their region of origin using antibody responses:

$$\log\left(\frac{\pi_{i,j}}{\pi_{i,c}}\right) = \bar{T}_i \times \beta_j^T \tag{3}$$

Where $\pi_{i,j}$ and $\pi_{i,c}$ are the probabilities of individual *i* being a resident of location *j* and *c* respectively, and $\bar{T}_i$ is the vector of antigen ranks selected for the model in individual *i*, and *βj* is the vector parameters for the *jth* logit.

## 3 Results

### 3.1 Responses to *Plasmodium* antigens are predictive of an individual's region of residence

First, protein microarray data from all 479 serum samples were pre-processed and within-sample ranks ($\bar{T}$, Section 2.3) of the relative reactivity to all antigens were computed based on fluorescence intensity. These values were then used as predictors in random decision forest models that classified all 479 serum samples into their geographical region of origin (Choma District, Zamiba; Nchelenge District, Zambia; or Mutasa District, Zimbabwe). To fine-tune our models we measured variable importance and defined top antigens to be those those with the highest mean decrease in Gini index from a random forest model with all 1038 *P. falciparum* and *vivax* antigens as predictors. We evaluated models with the top 20, 30, 65, 130, and 260 antigens as predictors and found that all of these models had global out of bag (OOB) error rates below 10% (Fig 1) and global AUC values above 0.95 (Table A in S1 Text). Similar results were obtained by computing test error (Fig A in S1 Text) and AUC (Table B in S1 Text) from a $\frac{2}{3}, \frac{1}{3}$ test-training split of the data. Furthermore, using a random set of 20, 30, 65, 130, or 260 antigens as predictors produced consistently higher OOB error rates (Fig 1), reinforcing that the top 20–260 antigens were the best predictors of region of origin. Models that used fewer than the top 10 antigens as predictors had substantially higher cross-validated and OOB error rates (Table F in S1 Text), suggesting that information contained in the relative levels, or ranks, of the top 10–20 antigens contribute substantially to differentiating antibody responses across the three regions. Performance of the random forest models using the top 20, 30, 65, 130, and 260 antigens as predictors were highly comparable (Fig 1, Table A, Table F in S1 Text), suggesting that adding more than 20 top antigens did not substantially improve the accuracy of the models. These patterns were consistent for global OOB error rates, and across all three region-specific error rates. Since malaria transmission levels across these three regions spanned a wide range of historic and current transmission intensities (Table 1), it follows that the exposure of people in this sample to *P. falciparum* parasites also spanned high, intermediate and low levels, both historically and at the time of sampling. The ability of these random forest models to classify samples by region of origin with high accuracy suggests that antibody responses to a subset of these antigens could be markers of the intensity and timing of past *P. falciparum* exposure.

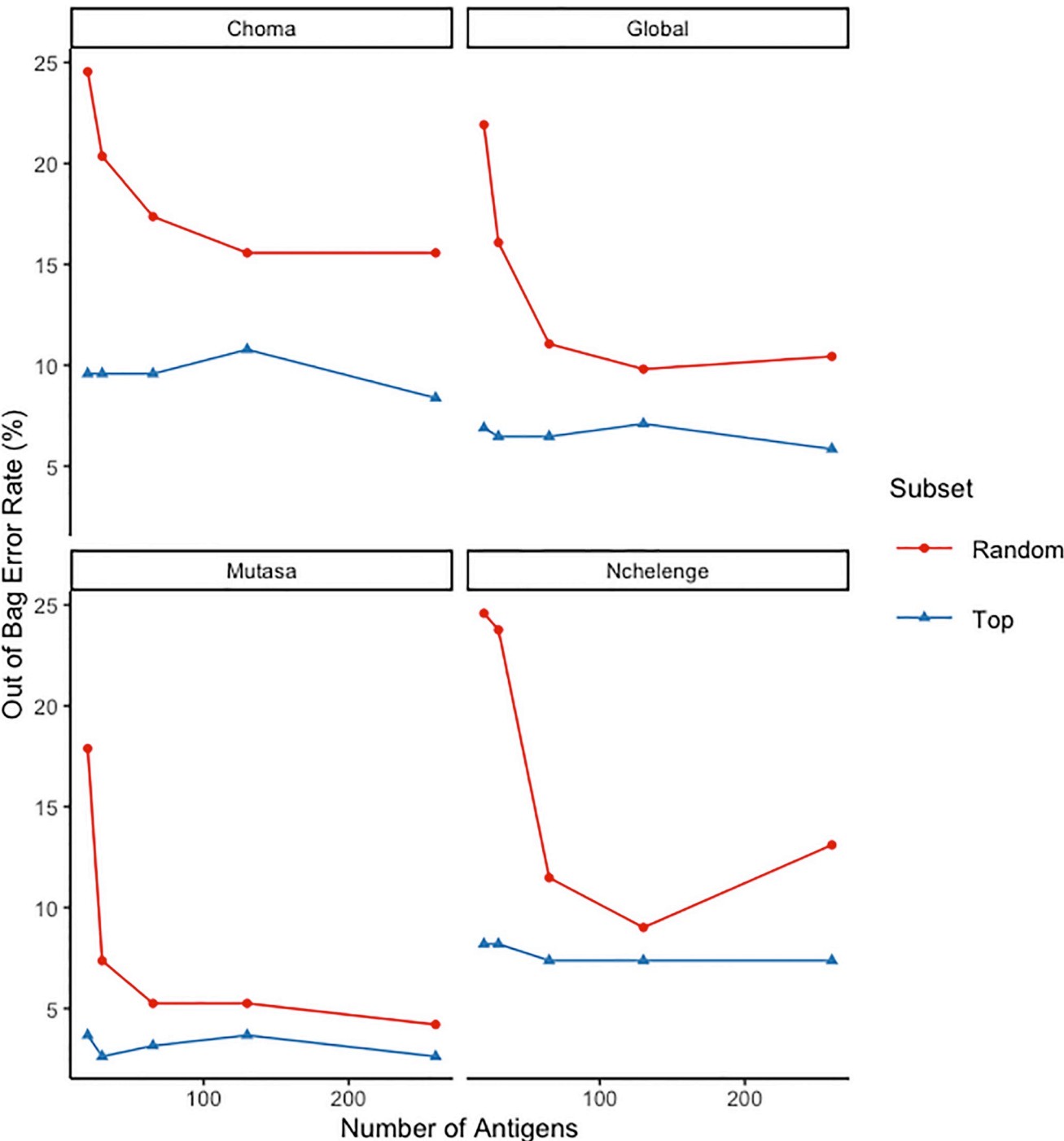

**Fig 1. Error rates for random forest models.** Out of bag error rate represented as a percentage for random forest models with 20, 30, 65, 130, and 260 antigens. Subsets of antigens are either the most important as determined by mean decrease in Gini index (Top, in blue), or a random subset (Random, in red). All models are fit to all 479 serum samples and out of bag error rates are reported for global, and category specific error rates (Choma, Nchlenege, or Mutasa).

## 3.2 Bioinformatic pipelines and analytic methods optimized for protein microarray data allow for an expanded analysis of biomarkers of past infection in adult populations

Elucidating markers of the timing and intensity of past exposure in adults is further complicated by uniformly high antibody responses to *P. falciparum* due to high numbers of re-exposure and re-infection events. However adults' infection histories span longer periods and therefore contain important information about historical transmission trends that enable longitudinal surveillance. Random forest models fit to subsets of samples revealed that the relative measures of antibody levels, or ranks, produced by the bioinformatic pipeline (Section 2) and analytic models that take advantage of interactions among antibodies, classified samples from adults (using an age cutoff of 5 years and of 15 years) into their region of origin with a global OOB error under 10% (Fig 2) and AUC values over 95% (Table C in S1 Text). The class-specific OOB error rates were much more variable for this subset analysis (Fig 2), this was likely due to the relatively small sample sizes of some subsets, particularly children under age 5 years. Although a formal test-training split was not carried out for the subset analysis, the similar values of AUC in Tables A and B in S1 Text obtained for classification with all samples suggest that AUC values in Table C in S1 Text would not change substantially under a similar test-training split. These results could indicate that differences in recent exposure lead to distinct antibody responses among adults and allowed the random forest models to classify samples into regions of different historical transmission patterns with high accuracy. However, given the duration of adults' exposure, it is also possible that longer lasting antibody responses were reflective of differences in historical exposure, and these are what enabled accurate classification among samples from adults. This second hypothesis was further supported by the model's high classification accuracy among RDT negative individuals (global OOB error under 10% Fig 2 and AUC over 95% Table C in S1 Text) who were unlikely to be harboring parasites in the weeks preceding sampling.

Using the relative levels of antibody response, or antigen ranks, from the bioinformatic pipeline (Section 2) as predictors, the random forest models outperformed polytomous, logistic regression models. The polytomous, logistic regression models did not include any interaction terms, that is terms that allow the effect of one factor to depend on levels other factors in the model. Random forest models allow for interactions among predictors without the need to pre-specify these relationships and explicitly take into account these interactions in variable importance measures [26]; several associative and predictive modeling studies have exploited interactions among predictors with the use of random forest models [31–33]. The difference between these models' performance was most extreme for models that used the top (determined by mean decrease in Gini index in the random forest model) 130 and 260 antigens (Table 2). These findings suggest that the interactions between ranked responses to antigens could help to distinguish the antibody responses to *Plasmodium* antigens of adults across zones of differing historical malaria transmission.

## 3.3 Most important antigens for classification reveal novel candidate biomarkers for the timing and intensity of past exposure to *P. falciparum*

Generally, there was little overlap between the biomarkers identified in previous studies aimed at identifying the intensity and timing of past exposure to *P. falciparum* and the most important antigens for classification via random forest identified here (as determined by mean decrease in Gini index). This suggests that analysis of these data revealed novel candidate biomarkers for the intensity and timing of past exposure to *P. falciparum*. The 30 most important antigens (Table 3) contain a a large proportion ($\frac{1}{3}$) of *P. vivax* antigens despite no evidence that

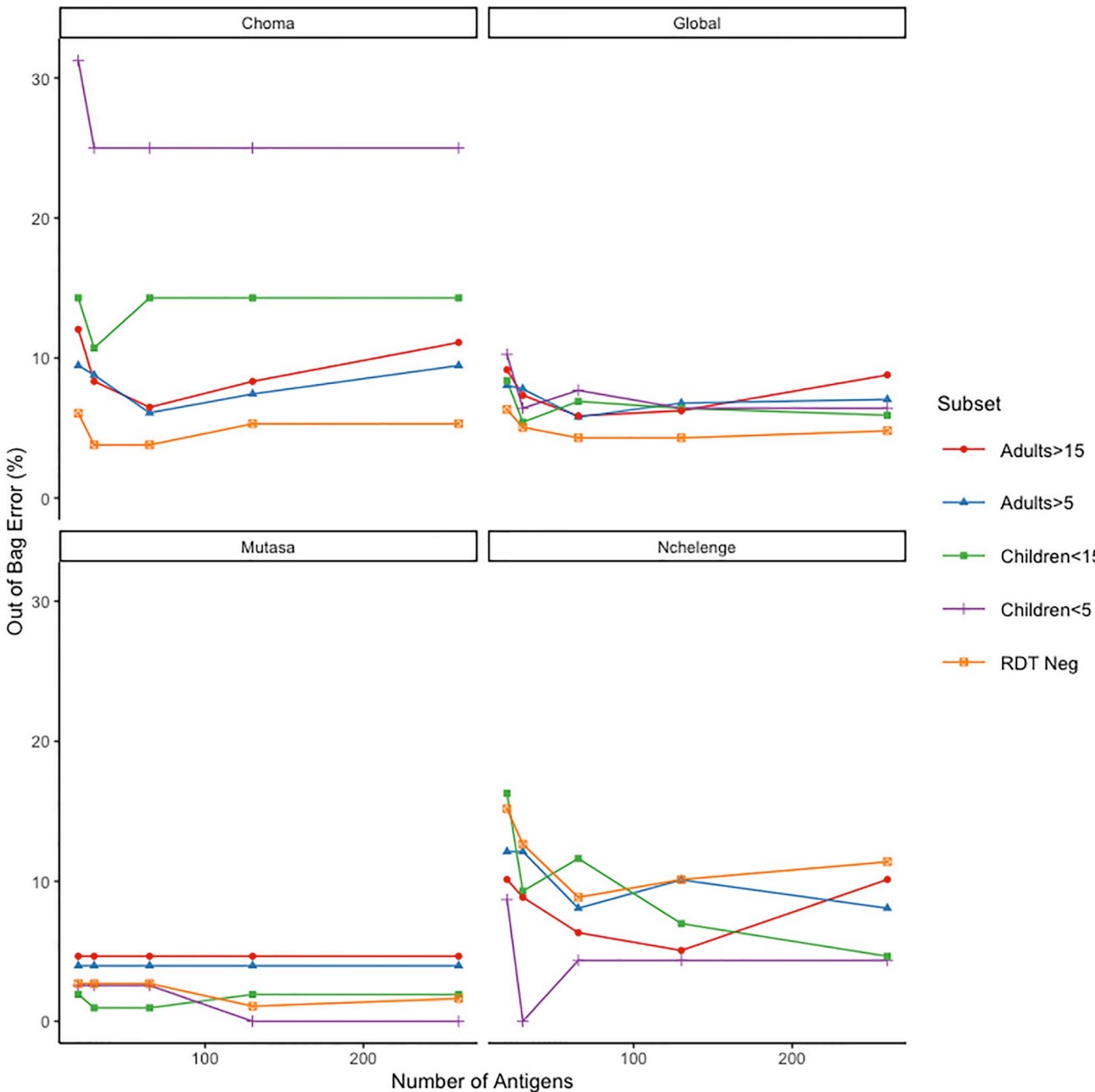

**Fig 2. Error rates for random forest models separated by age and RDT status.** Out of bag error rate represented as a percentage for random forest models with the most important 20, 30, 65, 130, and 260 antigens as determined by mean decrease in Gini index. Random forest models are run on five subsets of samples: children under 5 (n = 78), adults over 5 (n = 398), children under 15 (n = 203), adults over 15 (n = 273), and RDT negative (n = 396). Out of bag error rates are reported for global, and category specific error rates (Choma, Nchlenege, or Mutasa).

*P. vivax* circulated in any of the three study areas, although most of these proteins were designated as conserved suggesting they contained at least a motif of amino acids that is present across *Plasmodium* species, which may explain a certain level of cross-reactivity with antibodies generated after *P. falciparum* infection. Additionally, random forest models with only the top *P. falciparum* antigens (excluding *P. vivax* antigens as possible predictors) had similar

**Table 2. Results of polytomous regression models.** Area under the curve (AUC) and Akaike Information Criterion (AIC) [34] for polytomous, logistic regression models. Lower AIC values correspond higher estimates of a model's prediction accuracy. AUC is computed based on a $\frac{2}{3}, \frac{1}{3}$ train-test split of samples. The AUC reported is a multi-class AUC computed with the macro-average method as described in Hand and Till [30].

| Number of Antigens | 20 | 30 | 65 | 130 | 260 |
|---|---|---|---|---|---|
| AUC | 0.95 | 0.91 | 0.94 | 0.84 | 0.69 |
| AIC | 228 | 124 | 370 | 572 | 1145 |

OOB error to ones with the top *P. falciparum* and *vivax* antigens (Fig B and Table E in S1 Text). Two of the top 30 antigens, Plasmodium exported protein (PHISTc), and serine repeat antigen 4(SERA4) overlapped with published markers of exposure from Kobayashi et al. [17] and Helb et al. [15]. Additionally, certain proteins have been characterized in other studies related to human antibody response to *P. falciparum* infection including: clustered-asparagine-rich protein which has been shown to be recognized by human T-cells and antibodies in recently exposed individuals [35]; merzoite surface protien 3 (MSP3) which has been shown to generate antibody response during natural *P. falciparum* infection [36]; erythrocyte binding antigen 175 (EBA175), which has been shown to elicit antibody responses in children that are protective against clinical malaria [37]; and an erythrocyte membrane protein1, PfEMP1 (VAR), which appears as different proteins in the same family among the lists of biomarkers identified by Helb et al. [15] and Kobayashi et al. [17]. Aside from these, several proteins in the list of 30 most important antigens for classification (Table 3) could be novel biomarkers of the intensity and timing past exposure to *P. falciparum*, and considering the list of 65 most important antigens (Table D in S1 Text) reveals an expanded set of such candidate markers.

## 4 Discussion

Using protein microarrays, human antibody responses to *P. falciparum* and *P.vivax* antigens were characterized across serum samples collected from three regions in Zambia and Zimbabwe with historically high (Nchelenge District, Zamiba—parasite prevalence of 50%), intermediate (Mutasa District, Zimbabwe—parasite prevalence of 10%), and low (Choma District, Zambia—parasite prevalence of 1%) levels of malaria transmission [21, 22]. Previously, analysis performed using common pre-processing steps for protein microarray data found an association between responses to the 30 most reactive antigens and the region of origin of a sample, but only among children under 5 years [17]. We expanded this analysis, and overcame some obstacles associated with finding markers of infection history in older adults by considering responses to more antigens and by ensuring a minimal amount of technical variation was left in the data. We accomplished the goals of minimizing between-sample technical variation and expanding the list of candidate antigens with the application of a novel bioinformatic pipeline [23, 24]. Briefly, we obtained ranks based on fluorescence intensity of each antigen within each array that represent an optimal estimate [38] of the relative level of antibody response of an individual to a particular antigen, as compared to the response of that same individual to other *Plasmodium* antigens. Using the ranks of between 20 and 260 of the antigens measured on the protein microarrays as predictors in a random decision forest we classified samples into their region of origin with high accuracy. A polytomous, logistic regression model did not classify samples with such high accuracy, suggesting that differences in individuals' reactivity to these antigens, along with interactions among these predictors, was reflective of differences in past intensity and frequency of exposure to *P. falciparum*. The antigens that classified samples into their region of origin are candidate biomarkers for the timing and intensity of past

**Table 3. Novel candidate biomarkers of the timing and intensity of past infection.** The 30 most important *Plasmodium falciparum* and *P. vivax* antigens for classification via random forest identified according to the mean decrease in Gini index. Overlaps with previously published results from Helb et al. [15] and Kobayashi et al. [17] are shown.

| Importance Rank | Gene ID | Protein Name | Mean Decrease in Gini Index | Identified in Helb et al. [15] | Identified in Kobayashi et al. [17] |
|---|---|---|---|---|---|
| 1 | PVX_086275 | hypothetical protein, conserved | 6.6 | No | No |
| 2 | PVX_117340 | hypothetical protein, conserved | 5.2 | No | No |
| 3 | PF3D7_1032700 | conserved Plasmodium protein, unknown function | 4.5 | No | No |
| 4 | PF3D7_0207700 | serine repeat antigen 4 (SERA4) | 4.1 | No | Yes |
| 5 | PF3D7_1236100 | clustered-asparagine-rich protein | 3.9 | No | No |
| 6 | PF3D7_1015100 | conserved protein, unknown function | 3.7 | No | No |
| 7 | PF3D7_1149500 | ring-infected erythrocyte surface antigen 2, pseudogene (RESA2) | 3.7 | No | No |
| 8 | PVX_113590 | hypothetical protein, conserved | 3.5 | No | No |
| 9 | PF3D7_0108300 | conserved Plasmodium protein, unknown function | 3.5 | No | No |
| 10 | PF3D7_0801000 | Plasmodium exported protein (PHISTc), unknown function | 3.4 | Yes | Yes |
| 11 | PF3D7_0909500 | subpellicular microtubule protein 1, putative (SPM1) | 3.1 | No | No |
| 12 | PVX_087970 | cloroquine resistance associated protein Cg4, putative | 2.8 | No | No |
| 13 | PVX_088965 | hypothetical protein, conserved | 2.7 | No | No |
| 14 | PVX_092630 | hypothetical protein, conserved | 2.6 | No | No |
| 15 | PF3D7_0207800 | serine repeat antigen 3 (SERA3) | 2.3 | No | No |
| 16 | PVX_118070 | hypothetical protein, conserved | 2.2 | No | No |
| 17 | PVX_085220 | basic transcription factor 3b, putative | 2.2 | No | No |
| 18 | PF3D7_1328300 | conserved Plasmodium protein, unknown function | 2.2 | No | No |
| 19 | PF3D7_0103500 | conserved Plasmodium protein, unknown function | 2.1 | No | No |
| 20 | PVX_002550 | hypothetical protein, conserved | 2.1 | No | No |
| 21 | PF3D7_1343100 | conserved Plasmodium protein, unknown function | 2.1 | No | No |
| 22 | PF3D7_1150400 | erythrocyte membrane protein 1, PfEMP1 (VAR) | 1.9 | No | No |
| 23 | | MSP3, 0.1mg/mL | 1.9 | No | No |
| 24 | PF3D7_1311800 | M1-family alanyl aminopeptidase (M1AAP) | 1.9 | No | No |
| 25 | | EBA175, 0.1mg/mL | 1.8 | No | No |
| 26 | PF3D7_1134500 | alpha/beta hydrolase, putative | 1.8 | No | No |
| 27 | PVX_113500 | hypothetical protein, conserved | 1.8 | No | No |
| 28 | PF3D7_1014600 | transcriptional coactivator ADA2 (ADA2) | 1.7 | No | No |
| 29 | PF3D7_0817300 | asparagine-rich antigen | 1.6 | No | No |
| 30 | PF3D7_0702400 | conserved Plasmodium protein, unknown function | 1.6 | No | No |

exposure to *P. falciparum*, and could overcome challenges associated with current surveillance methods.

Although the analysis we performed identified candidate biomarkers for the timing and intensity of past exposure, there are several additional studies and steps that need to be carried out before any biomarkers can be validated and used in the field for surveillance. A longitudinal study with detailed information about participant's infection status over time, possibly

spanning decades, would be required to directly associate differences in antibody response to these antigens with differences in infection history. Moreover, given the geographical heterogeneity in the genomes of *P. falciparum* parasites, it is possible that the antibody responses in the study participants in Zambia and Zimbabwe are not easily generalized to other populations; a more extensive geographical sample across sub-Saharan Africa is required. Furthermore, protein microarray data, even with careful pre-processing, remains prone to high levels of technical variation and does not allow analysts to directly measure antibody titers. Therefore, a lower throughput assay that does enable such quantities to be estimated, like a multiplex bead array or an enzyme-linked immunosorbent assay (ELISA), should be performed with the candidate antigens identified here to further validate their potential as biomarkers for the timing and intensity of past exposure to *P. falciparum*. For many of these antigens particularly those identified as hypothetical proteins or those with unknown function, additional investigation into the biochemical composition and identity of these antigens is also warranted. Furthermore, it is possible that other factors that contribute to regional differences in antibody responses such as human and parasite genetic diversity could driving some of the results observed in this analysis. Finally, some samples from the low transmission setting (Choma District, Zambia) were longitudinal measurements and this was not taken account in our analysis, however, this was a minority of samples (65 of the 166 samples in this area).

The number of *P. vivax* antigens that were among the most important for classifying samples into their region of origin, 10 of the top 30 and 25 of the top 65, is surprising given that cases of *P. vivax* have not been and are not currently documented in any of the sampling regions. Many of the *P. vivax* antigens in list of top 30 (Table 3) or top 65 antigens (Table D in S1 Text), were conserved proteins, meaning they contained at least a motif that is conserved across *Plasmodium* species. Therefore, this finding could be the result of antibodies generated in response to *P. falciparum* exposure that cross-react with *P. vivax* antigens. However, further investigation into the possible utility of including *P. vivax* antigens in the search for markers of the timing and intensity of past *P. falciparum* exposure may be worthwhile.

Generally, our analytic methods identified different antigens of interest than those identified by Kobayashi et al. (2019) [17] and although our analysis contained an additional 34 additional samples from Mutasa District, Zimbabwe, a sensitivity analysis without those 34 samples revealed highly similar results (see Fig C in S1 Text). While we did not directly compare the performance of a random forest classifier using ranks from all stages of our bioinformatic pipeline (Pre-processing pipeline [23] and Bayesian model [24]), Bérubé et al. (2023) have shown that the ranks of antigens from this same dataset change substantially at the two stages of the bioinformatic pipeline implying that the downstream inference associated with these two different sets of ranks would also change. Loss-function optimal ranking methods, which have been shown to outperform other methods [38], cannot be applied to the estimates from the pre-processing pipeline since they rely on the use of estimates in the form of full probability distributions, such as those obtained from a Bayesian estimation procedure. Finally, it is not possible to assess the importance of the random forest classifier, or a similar classification approach, since a classification method of some sort is needed to directly answer our question of interest (identifying the antigens that could be used as markers of the timing and intensity of past exposure); observing the ranks alone is not sufficient to answer this question. Therefore, a random forest classifier that uses the loss-function optimal ranks as predictors, as presented in this paper, is an appropriate method to identify markers of the timing and intensity of past exposure to *P. falciparum*. That said, the use of this method in no way invalidates existing work on markers of recent exposure published with other pre-processing and analytic pipelines such as the one in Helb et al. [15]; our work is an extension of existing publications.

There are several possible reasons besides the difference in analytic methods for the lack of overlap between antigens identified in our analysis and those previously published. There was a high overlap in the number of antigens screened across the two studies and of the antigens identified in Helb et al. [15] and only 3 (Dihydrolipoamide acyltransferase component E2; Plasmodium exported protein, PHISTb; and Secretory complex protein 62) were absent from the antigens screened in our analysis. However, the study population in Helb et al. [15] differed from the study population in this analysis in ways that are known to be associated with differences in antibody response, namely, the population analyzed in Helb et al. [15] resided areas of moderate to intense malaria transmission while our study areas included a very low transmission region and an area with resurgent malaria after successful control. Furthermore, the population in Helb et al. [15] consisted of young children while our analysis included responses from individuals aged 0 to 86 years old.

Although further studies are required to validate the candidate biomarkers we have identified in this study, narrowing down a set of possible antigens to target is an important first step toward eventually integrating serology into routine malaria surveillance. Ultimately, validated markers of the timing and intensity of past exposure to *P. falciparum* could enable analysts to accurately assess the burden of malaria over long periods of time with the ease of a cross-sectional sample. This, in turn could facilitate targeting effective interventions in low and high transmission settings and accurately evaluating the effectiveness of such interventions.

## Supporting information

**S1 Text.**
(PDF)

## Acknowledgments

The authors are grateful to Philip Felgner and Huw Davies for sharing data relating to their *Plasmodium falciparum* and *P. vivax* antibody arrays.

## Author Contributions

**Conceptualization:** Sophie Bérubé, Douglas E. Norris, Ingo Ruczinski, William J. Moss, Amy Wesolowski, Thomas A. Louis.

**Data curation:** Tamaki Kobayashi.

**Formal analysis:** Sophie Bérubé, Thomas A. Louis.

**Funding acquisition:** William J. Moss, Amy Wesolowski.

**Investigation:** Sophie Bérubé.

**Methodology:** Sophie Bérubé, Tamaki Kobayashi, Ingo Ruczinski, Amy Wesolowski, Thomas A. Louis.

**Supervision:** Thomas A. Louis.

**Writing – original draft:** Sophie Bérubé, Amy Wesolowski, Thomas A. Louis.

**Writing – review & editing:** Sophie Bérubé, Tamaki Kobayashi, Douglas E. Norris, Ingo Ruczinski, William J. Moss, Amy Wesolowski, Thomas A. Louis.

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
