## [Decision Letter · Decision Letter 0]

27 Feb 2023

PGPH-D-22-01964

A Random Forest Classifier Uses Antibody Responses to Plasmodium Antigens to Reveal Candidate Biomarkers of the Intensity and Timing of Past Exposure to Plasmodium falciparum

Dear Dr. Berube

Thank you for submitting your manuscript to PLOS Global Public Health. After careful consideration, we feel that it has merit but does not fully meet PLOS Global Public Health’s publication criteria as it currently stands. Therefore, we invite you to submit a revised version of the manuscript that addresses the points raised during the review process.

We look forward to receiving your revised manuscript.

Kind regards,

Janelisa Musaya, PhD

Academic Editor

Journal Requirements:

2. Please send a completed 'Competing Interests' statement, including any COIs declared by your co-authors. If you have no competing interests to declare, please state "The authors have declared that no competing interests exist". Otherwise please declare all competing interests beginning with the statement "I have read the journal's policy and the authors of this manuscript have the following competing interests:"

4. We ask that a manuscript source file is provided at Revision. Please upload your manuscript file as a .doc, .docx, .rtf or .tex.

5. Please provide separate figure files in .tif or .eps format.

6. Please note that your Data Availability Statement is currently missing the repository name and a direct link to access each database. If your manuscript is accepted for publication, you will be asked to provide these details on a very short timeline. We therefore suggest that you provide this information now, though we will not hold up the peer review process if you are unable.

7. We have noticed that you have uploaded Supporting Information files, but you have not included a list of legends. Please add a full list of legends for your Supporting Information files after the references list. 

Additional Editor Comments (if provided):

Reviewers' comments:

Reviewer's Responses to Questions

**Comments to the Author**

1. Does this manuscript meet PLOS Global Public Health’s publication criteria? Is the manuscript technically sound, and do the data support the conclusions? The manuscript must describe methodologically and ethically rigorous research with conclusions that are appropriately drawn based on the data presented.

Reviewer #1: Partly

Reviewer #2: Yes

2. Has the statistical analysis been performed appropriately and rigorously?

Reviewer #1: I don't know

Reviewer #2: Yes

3. Have the authors made all data underlying the findings in their manuscript fully available (please refer to the Data Availability Statement at the start of the manuscript PDF file)?

Reviewer #1: No

Reviewer #2: Yes

4. Is the manuscript presented in an intelligible fashion and written in standard English?

Reviewer #1: Yes

Reviewer #2: Yes

5. Review Comments to the Author

Reviewer #1: The authors’ main claims of the paper are use of 2 new techniques to successfully classify samples of antibody responses to malaria protein microarrays by location of origin which the authors state is a proxy for malaria transmission.

- Bioinformatic preprocessing (in biorxiv) and ranking model

- Random forest

The authors compared to their previous work (Kobayashi et al), some samples of which appear to have overlap, in which only antibody responses in the children <5 y were predictive of location. However, there are open questions:

- The previous work had fewer samples. Is better success in classification due to the larger sample size?

- It’s not clear whether all three techniques are needed for the successful model, and this could be tested by testing each of the new techniques separately. Furthermore, consider whether the title of the paper should include the ranking model as part of the innovation that led to successful classification.

- The reason for improved success is suggested to be (lines 254-255) random forest analyses include consideration of “interactions between the antibody responses” – I don’t know one way or the other whether RF does that – I suggest biostatistician review and/or more clarification by the authors because it is stated as “fact”.

- Besides their previous work, they also compared to Helb et al, and there is little overlap in proteins identified between the current study and Kobayashi and Helb studies, leaving some questions:

o Were the same proteins tested? And were Helb proteins and Kobayashi proteins similar? And, what is the significance of proteins identified in one study but not the others (is one study more valid, or are all the studies valid? Furthermore, what would results be if the methods were applied to the Helb dataset?

o It would be good to know which of the Kobayashi samples are also in the current study.

Overall, I would have great enthusiasm for this work if the contributions of the different components of the models (ranking, RF, logistic regression) could be tested separately to better understand what makes everything work so well, and also more pontification on why new proteins are found with each analysis with little overlap.

Minor points

Line 106 – seems to be a discussion point (also is a major point that I am not sure is supported by the manuscript, see comment about lines 254-255 above)

3.1 and Table 1. make clear whether the 479 samples are from 479 individual people or from some people measured more than once, if some samples are from people measured at multiple timepoints, address this.

Line 129 – decline in transmission in Mutasa District – when in 2015 were the samples collected (before or after the decline?)

Verb tense varies throughout materials and methods (115, “were probed”; 135 “is fed”)

Verb tense varies throughout results (205 “found that”; 233 “classify samples”)

Line 145 – P = 1038?

Data model (equation 1) not fully explained, but there is a pre-print. Consider either fully explaining it (all Greek letters) or omitting it. My understanding from the manuscript is that the preprocessing can refer to the preprint but that the ranking method is specific to this manuscript.

Throughout: fluorescence intensity (not fluorescent) is more commonly used

Suggest including additional reasons that antibody levels identify region (human genetics, parasite diversity)

221 “random forest models TO classify” (word missing

The polytomous logistic regression models should be included in the methods

Data availability - I see location of code for random forest but not location of data (protein microarray data), if in a repository it should be listed in the manuscript.

Reviewer #2: The authors present a study on malaria surveillance through measuring the levels of antibodies in response to 1000 antigens of P. falciparum and P. vivax. By building a Random Forest model, the authors seek to answer which antibodies are markers for past exposure to the parasite.

The model classifies samples from different regions using the antibody response (fluorescent intensity) of 479 sera (collected around 2015) as predictors.

As different regions have different rates of transmission, therefore, the classification is indicative of past exposure. This is a bold assumption for machine learning modeling. In fact, I believe this is the first time artificial intelligence has been used for microarray data analysis, which makes the work quite interesting, allowing the high throughput analysis with hundreds of antigens.

I suggest simplifying the title to better hold the reader, using "artificial intelligence" instead of “random forest” since few readers have this knowledge.

The authors present in the Discussion section some limitations regarding the present study and point out future directions, such as the necessity of longitudinal studies and future validation with lower throughput techniques (multiplex beads assay or ELISA). The authors also comment on the fact that many P. vivax antigens appear among the most recognized since this parasite does not circulate in these countries. However, it is known that many antigens have conserved regions between different species of Plasmodium, which can then be recognized by the immune system.

It is very interesting that among the most recognized antigens, important proteins such as MSP1 and MSP2 did not appear. However, the presence of other MSPs and PfEMP1 are good indications that the results are robust. Also noteworthy is the large amount of hypothetical proteins and of unknown function.

I just suggest correcting minor typos, and reporting the best model performance results and best predictors in the abstract. Perhaps showing the best model performance metrics (accuracy, sensitivity, AUC, etc.) in the abstract can make the model more convincing.

6. PLOS authors have the option to publish the peer review history of their article (what does this mean?). If published, this will include your full peer review and any attached files.

**Do you want your identity to be public for this peer review?** For information about this choice, including consent withdrawal, please see our Privacy Policy.

Reviewer #1: No

Reviewer #2: No

---

## [Decision Letter · Decision Letter 1]

29 Jun 2023

Novel Bioinformatic Methods and Machine Learning Approaches Reveal Candidate Biomarkers of the Intensity and Timing of Past Exposure to Plasmodium falciparum

PGPH-D-22-01964R1

Dear Sophie Berube,

We are pleased to inform you that your manuscript 'Novel Bioinformatic Methods and Machine Learning Approaches Reveal Candidate Biomarkers of the Intensity and Timing of Past Exposure to Plasmodium falciparum' has been provisionally accepted for publication in PLOS Global Public Health.

Best regards,

Janelisa Musaya, PhD

Academic Editor

Reviewer Comments (if any, and for reference):

Reviewer's Responses to Questions

**Comments to the Author**

1. If the authors have adequately addressed your comments raised in a previous round of review and you feel that this manuscript is now acceptable for publication, you may indicate that here to bypass the “Comments to the Author” section, enter your conflict of interest statement in the “Confidential to Editor” section, and submit your "Accept" recommendation.

Reviewer #1: All comments have been addressed

Reviewer #2: All comments have been addressed

2. Does this manuscript meet PLOS Global Public Health’s publication criteria? Is the manuscript technically sound, and do the data support the conclusions? The manuscript must describe methodologically and ethically rigorous research with conclusions that are appropriately drawn based on the data presented.

Reviewer #1: Yes

Reviewer #2: Yes

3. Has the statistical analysis been performed appropriately and rigorously?

Reviewer #1: I don't know

Reviewer #2: Yes

4. Have the authors made all data underlying the findings in their manuscript fully available (please refer to the Data Availability Statement at the start of the manuscript PDF file)?

Reviewer #1: Yes

Reviewer #2: Yes

5. Is the manuscript presented in an intelligible fashion and written in standard English?

Reviewer #1: Yes

Reviewer #2: Yes

6. Review Comments to the Author

Reviewer #1: The reviewers addressed my previous questions and comments. Additional comments:

Evaluation with and without inclusion of P. vivax proteins – if the P. vivax proteins are included or excluded, do the same P. falciparum proteins end up as top candidate biomarkers? if largely "yes," I feel this validates the method because if the antigen is important, it should be identified as important, regardless of presence or absence of the vivax proteins. Alternatively, perhaps there is some fluctuation in the variable importance of the P. falciparum antigens due to interactions? Since you have broached this by including and excluding the P.vivax antigens, I think a little more clarification would be helpful.

Also, for the P. vivax proteins, rather than inferring that if the name contains "conserved" that the proteins are conserved, were any types of sequence analysis between Pv and Pf done? (a quick look by eye, or a pairwise alignment?) it would be helpful to know if the Pv antigens might be cross-reactive with antibodies elicited by Pf. Also, are any of the Pv. Antigens identified actually homologous to top Pf antigens that were identified?

For the figures and tables, I think some of the supplementary figures and tables are not listed in the order in which they are mentioned in the text.

For Figure S2, suggest highlighting the difference between this figure and figure 2 by including the number of proteins that were considered (denominator), which would make it more clear that Figure S2 considered only the falciparum proteins and Figure 2 considered all the proteins. Also, suggest sticking with the same colors as in Figure 2. Same for Table S5 and all of the corresponding tables – it would help to somehow signify how the tables different (Pf + Pv vs Pf only by either saying “pv not included here” in the supplementary tables or listing the number of antigens considered so that people could quickly see the supplementary tables have half as many.

Page 6 of supplement, title says “Table 4” but this should be “Table S4”

Reviewer #2: All my comments have been addressed

7. PLOS authors have the option to publish the peer review history of their article (what does this mean?). If published, this will include your full peer review and any attached files.

**Do you want your identity to be public for this peer review?** For information about this choice, including consent withdrawal, please see our Privacy Policy.

Reviewer #1: No

Reviewer #2: No
